# OpenReview forum: "One Pic is All it Takes: Poisoning Visual Document Retrieval Augmented Generation with a Single Image"
_TMLR — Accepted by TMLR_

### Review · Reviewer_xySe · 2025-12-22

**Summary Of Contributions:**

Firstly, I’m sorry for my late review. It was an unusual week for me.

The retrieval augmented generation (RAG) is a VLM text generation setting in which the model is conditioned on a knowledge base to generate more grounded responses. In the scenario that this paper considers, the knowledge base consists of a number of images, and the VLM uses the top-k images that are closest to the user query in a joint embedding space.

What this paper shows is that this setting can be disrupted by both white-box and black-box attacks by injecting only a single image. The paper presents two types of attacks. In the first one, the attack aims to spread misinformation (or disrupt the system) on a targeted group of user queries. In the second, the disruption happens regardless of the user query (i.e., universal). The attack is based on updating the injected image with projected gradient descent (PGD) to maximize both its retrieval probability and the generation of malicious output when conditioned on that image. The attacker knows the embedding model in white-box attacks, and thus, can do PGD on it, whereas in black-box attacks, the attacker relies on surrogate models to compute gradients. Lastly, the paper employs several well-known defense techniques, though these are not specialized for the RAG attacks defined in the paper.

There are extensive experiments for each of these scenarios and their subcases with multiple VLMs, showing that these attacks are indeed realizable in some visual document retrieval datasets.

**Audience:**

Yes

**Audience Explanation:**

The topic would definitely be of interest to some people in the machine learning community, more specifically, to those focusing on adversarial attacks (and defenses) on VLMs, and to people who study ethics in the context of large models.

**Broader Impact Concerns:**

Although there's an ethics and societal impact section in the appendix, I think this should be moved to the main section, given the topic of the paper. It's a study that can be easily read, and has the potential to be read by broader audience. You might even consider expanding the discussion on white-box and black-box scenarios, but I leave the decision to you.

**Claims And Evidence:**

Yes

**Claims Explanation:**

The claims are clearly supported by extensive experiments on multiple datasets with several models. Metrics make sense.

**Requested Changes:**

I think the paper is already in a very good shape. It motivates the problem and clearly sets out the setting. Methods are well-defined. I didn't catch any ambiguity in the text. Below are some small things you might consider for future versions.

- I would omit the human head depiction in Figure 1 for VLMs—might be confused with some user-related things.
- It’s a small thing but, in Equation 2, why do we subtract from 1? Won’t $\Sigma_{i=1}^{\mathcal{Q}^{-}} S(E(q_i^{-}), E(I’)) - \Sigma_{i=1}^{\mathcal{Q}^{+}} S(E(q_i^{+}), E(I’))$ look simpler?

This is a bit more important than the above two:
- It’s somewhat surprising to me that prompt-based black-box attacks are quite successful, considering that we don’t have any access to the model, which is a bit concerning (not for the paper, of course, but as a broader impact) since this is rather a likely scenario. This also makes me question the overall efficiency of gradient-based attacks; they are probably overfit to the target model, and these prompt-based attacks seem to be more generalized. This might be worth mentioning earlier in the text, maybe in the introduction. I'm not saying that this has to be included, but you might consider whether that's worth it.

---

### Review · Reviewer_DS4G · 2025-12-25

**Summary Of Contributions:**

This paper studies poisoning attacks on VD-RAG systems and shows that injecting a single malicious image in the knowledge base can be enough to compromise both retrieval and generation. The attacks can be used to generate two malicious behaviors -- spreading targeted misinformation or DOS.

It glues together existing methods of creating malicious images (PGD with a multi-objective loss) and applies it to attack retrieval and generation, which is novel.

The work includes meaningful negative results (e.g., limited transfer of attacks and black-box effectiveness in some settings) and thorough ablations that clarify when and why attacks succeed. It explores a wide range of attack scenarios (white-box, black-box), defenses, datasets, and model combinations, which is a huge strength of this work. However, the study is limited primarily to small VLLMs with under 4B parameters.

**Audience:**

Yes

**Audience Explanation:**

Yes, the paper shows that VD RAG systems are vulnerable to single image poisoning, highly relevant read given that a lot of RAG systems today are built on PDF systems. The methodological contribution of adapting MO-PGD framework specifically for RAG pipelines is also interesting. Plus, backing most claims with extensive empirical ablations make it a good read.

**Broader Impact Concerns:**

It would be helpful to add one since this work exposes techniques to poison RAG systems. This should include ethical considerations and potential misuse.

Several other TMLR publications that showcase attacks include such a statement -- https://openreview.net/pdf?id=Lvy5MjyTh3, https://openreview.net/pdf?id=MlUP5Euj6S, https://openreview.net/pdf?id=2ekgTdBOZo

**Claims And Evidence:**

Yes

**Claims Explanation:**

Yes. The main claim is that VD RAG systems are vulnerable and the authors have done a great job at putting together several ablations on what works when. They have also added negative results to help the readers understand when the attacks and defenses fail.

Attack settings include targeted and universal poisoning, with variations across white-box and black-box assumptions, showing that attacks are highly effective in white-box and targeted settings but degrade significantly in black-box and universal cases.

Defenses include knowledge expansion, VLM-as-a-judge, and query paraphrasing, showing they are only partially effective and can be bypassed by adaptive attacks.

All of which are supported by experiments on prominent embedding models and small scale VLLMs.

**Requested Changes:**

It would be nice to call out that these experiments and claims are limited to small-scale RAG systems as the VLLMs they deal with are all small scale VLLMs and not all VLLMs. Alternatively, it would be helpful to include some experiments with more powerful VLLMs of varying sizes and see if they also have similar robustness issues.

---

### Review · Reviewer_TsaN · 2026-01-06

**Summary Of Contributions:**

This paper analyses knowledge base (KB) poisoning in visual document RAG (VD-RAG), where the KB consists of document screenshots, and retrieval is done using multimodal text-image embedding models. Specifically, authors analyse poisoning under \emph{retrieval condition} (the poisoned document is retrieved for attacker-chosen queries); and \emph{generation condition} (when in context, the VLM produces attacker-chosen content). They introduce two attack objectives with a single injected image: a \emph{targeted disinformation} objective affecting a single query or a small set of related queries; and an \emph{universal denial-of-service} (DoS) objective intended to affect any query. In their approach, authors adapt projected gradient descent (PGD) to multi-objective optimisation (MO-PGD) that combines a retrieval loss (increasing similarity to target queries while decreasing similarity to non-target queries) and a generation loss (cross-entropy for a target answer). Authors evaluate on ViDoRe datasets (V1-AI and V2-ESG) with three retrievers (CLIP-ViT-L, ColPali-v1.3, and GME-Qwen2-VL-2B) and three open-weights VLMs (SmolVLM, Qwen2.5-VL-3B, and InternVL3-2B) under both white-box and several black-box variants (prompt-based image generation, transfer, and model ensembles); they also evaluate common defence mechanisms (knowledge expansion, VLM-as-a-judge, and query paraphrasing) and via several analyses and ablations discuss why CLIP seems to be significantly more vulnerable to the universal objective than ColPali/GME.
This paper is both extremely timely and practically relevant -- VD-RAG is increasingly deployed in industry, and KB poisoning has mainly been explored in text-only settings. The experimental insights are very interesting, e.g., due to the clear decomposition of the poisoning success into retrieval+generation, the exhaustive sweeps (datasets, retrievers, VLMs, attack variants, defence mechanisms..), and the simple but effective MO-PGD objective.

**Audience:**

Yes

**Audience Explanation:**

Anyone interested in VLMs and security/safety will likely find this paper interesting.

**Broader Impact Concerns:**

I don't think this paper raises ethical concerns.

**Claims And Evidence:**

Yes

**Claims Explanation:**

The paper shows that VD-RAG systems can be poisoned with a single image, especially under white-box settings, and that the generation component can be very sensitive once a malicious image is in context. The paper also shows that universal poisoning is significantly harder and largely limited to CLIP-like retrieval modules.

**Requested Changes:**

In the evaluations, it would be extremely valuable to have more decision-based success measures (vs embedding similarity).

---

> ### Author Response · Authors · 2026-01-07
>
> Thank you for your feedback. Below we present our response to the point raised.
>
> > In the evaluations, it would be extremely valuable to have more decision-based success measures (vs embedding similarity).
>
> In preliminary evaluations (not shown in the paper) we considered a stricter metric for assessing the attack success w.r.t. the generation condition, which measures whether the VLM generated the malicious target response verbatim.
> We decided to use the embedding similarity-based metrics since they give a more nuanced view of the attack success (e.g., if the generated output is very close to the target response but not exactly the same, the embedding similarity could still indicate a successful attack, while the verbatim score will be zero).
>
> An alternative metric could be querying an LLM to judge whether the generated response is close enough to the target response.
> While this metric is also nuanced, it might induce biases based on the LLM used as a judge and its prompt.

---

### Review · Reviewer_CWyY · 2026-01-19

**Summary Of Contributions:**

The paper investigates the vulnerability of Visual Document Retrieval Augmented Generation (VD-RAG) systems to data poisoning attacks. Unlike text-based RAG, VD-RAG retrieves document screenshots directly. The authors demonstrate that injecting a single malicious image into the knowledge base is sufficient to compromise the system.

Key contributions and findings include:

- Defining two attack objectives for VD-RAG: Targeted Disinformation (affecting specific queries) and Universal Denial-of-Service (affecting all queries).

- Adapting Projected Gradient Descent to a Multi-Objective optimization framework (MO-PGD) that simultaneously optimizes an image for retrieval (high similarity to queries) and generation (inducing specific VLM responses).

- Testing the attacks across various settings (white-box vs. black-box), datasets (ViDoRe-V1-AI, ViDoRe-V2-ESG), and modern retrieval/generation models (CLIP, ColPali, GME, SmolVLM, Qwen2.5-VL).

- VD-RAG is highly vulnerable to white-box poisoning.

- Older embeddings (CLIP) are susceptible to universal attacks due to the modality gap, while newer interactions (ColPali) are more robust to universal attacks but still vulnerable to targeted ones.

- Common defenses (Knowledge Expansion, VLM-as-a-judge, Paraphrasing) are largely ineffective against adaptive attacks.

- Black-box attacks generally struggle, with the exception of prompt-based generation attacks in targeted settings.

**Additional Comments:**

The paper is well-written and the experimental design is robust. The decomposition of the attack into "Retrieval Condition" and "Generation Condition" makes the analysis very clear.

**Audience:**

Yes

**Audience Explanation:**

RAG systems are currently ubiquitous in deployed LLM applications. As the field moves toward Multimodal/Visual RAG to handle complex documents (PDFs with charts, figures), understanding the security surface of these new architectures is critical. This paper is timely as it highlights a specific vulnerability (visual poisoning) that does not exist in text-only pipelines, directly interesting researchers in adversarial ML, multimodal learning, and safety/security.

**Broader Impact Concerns:**

The paper exposes a method to poison RAG systems, which could potentially be misused. However, the authors have included a "Ethics & Societal Impact" section (Section 10) acknowledging this risk and framing the work as a necessary step for defense development. I believe this is sufficient and standard for adversarial ML papers. No additional changes are required here.

**Claims And Evidence:**

Yes

**Claims Explanation:**

- Tables 1-4 clearly quantify the drop in performance (or success of attack) across different metrics when a poisoned image is introduced.
- The paper meticulously dissects why certain models are robust. For instance, the UMAP visualizations (Figure 6) and ColPali ablations (Table 7) convincingly argue that the robustness of ColPali to universal attacks stems from its late-interaction mechanism and lack of a distinct modality gap, compared to CLIP.
- The evaluation of defenses is not just a checkbox; the authors perform adaptive attacks (e.g., training against the judge), providing a realistic assessment of security risks.

**Requested Changes:**

- The prompt-based attacks (using Gemini/GPT to generate images) are surprisingly effective in targeted settings compared to other black-box methods. Looking at Figure 4, the generated images contain visible text (e.g., "Manually match each marker..."). It would strengthen the paper to explicitly discuss if these attacks are successful primarily because they act as "typographic attacks"—exploiting the VLM's OCR capabilities—rather than adversarial perturbations in the pixel space. A brief comment on this distinction in Section 4 would provide valuable insight into why this baseline works.

- The victim VLMs used (SmolVLM, Qwen2.5-VL-3B, InternVL3-2B) are relatively small (<4B parameters). While I understand the compute constraints for gradient-based attacks, please add a brief discussion or disclaimer in the Limitations section regarding whether you expect larger models (e.g., GPT-4V, Gemini 1.5 Pro) to exhibit similar vulnerabilities, particularly regarding the "generation condition."

---

> ### Author Response · Authors · 2026-01-20
>
> Thank you for the feedback.
>
> > The prompt-based attacks (using Gemini/GPT to generate images) are surprisingly effective in targeted settings compared to other black-box methods. Looking at Figure 4, the generated images contain visible text (e.g., "Manually match each marker..."). It would strengthen the paper to explicitly discuss if these attacks are successful primarily because they act as "typographic attacks"—exploiting the VLM's OCR capabilities—rather than adversarial perturbations in the pixel space. A brief comment on this distinction in Section 4 would provide valuable insight into why this baseline works.
>
> We agree with the reviewer's remark that the text in the generated image is most likely the reason why the prompt-based attacks are successful. As a black-box attack, this attack does not have access to the model weights and hence cannot introduce meaningful adversarial perturbations to an image in the pixel space. Interestingly, a human tasked with the same adversarial goal will most likely rely on injecting text in the image to achieve both the retrieval and generation conditions.
>
> We have revised the paper to add the following comment on this while discussing the partial success of the prompt-based attacks in Section 4 (Setting I: One Query):
>
> *"We attribute this partial success to the textual (typographic) elements in those generated images (see examples of successful Prompt-based attacks in Appendix A), exploiting the OCR capabilities of both the embedding model and the VLM."*
>
>
> > The victim VLMs used (SmolVLM, Qwen2.5-VL-3B, InternVL3-2B) are relatively small (<4B parameters). While I understand the compute constraints for gradient-based attacks, please add a brief discussion or disclaimer in the Limitations section regarding whether you expect larger models (e.g., GPT-4V, Gemini 1.5 Pro) to exhibit similar vulnerabilities, particularly regarding the "generation condition."
>
> While we do expect that the studied attacks would still work against larger open-weight models (we experimented with models ranging from 256M to 4B parameters with very similar results), generating successful white-box attacks against proprietary closed-weight models (e.g., GPT-4V, Gemini 1.5 Pro) is not possible as white-box attacks require access to gradients.
> For black-box prompt-based attacks, we have briefly experimented with providing those adversarial images along with the user query to GPT 5.2 using the web interface (i.e., assuming that the retrieval condition is passed, ASR-R=1) and the attack succeeds in most trials in generating the target answer. Note that we have not experimented with a closed-weight embedding model, as we are not aware of any existing closed-weight multi-modal embedding model.
>
> In the revised version, we elaborated on the existing discussion on larger models in the limitation section by adding the following statement:
>
> *"Nevertheless, we conjecture that our attacks will succeed against larger models, as experimenting with model sizes between 256M to 4B parameters (spanning more than one order of magnitude) yielded very similar vulnerability.
> We have also conducted our experiments on open-weight models only, as white-box attacks could only be evaluated against those. Investigating the vulnerability of proprietary closed-weight model to black-box attacks is an interesting direction for future work."*

---

### Author Response · Authors · 2026-01-07
**Paper Revision**

We have uploaded a revision of the paper including the following:

1. As suggested by reviewers xySe and DS4G, the "Ethics and Societal Impact" section is moved to the main paper body.
2. In response to reviewer DS4G, the "Limitations" section is moved to the main paper body.
3. As suggested by reviewer xySe, the head image is removed from Figure 1, to avoid confusion with the VD-RAG user.
4. As suggested by reviewer xySe, the subtraction from one is removed in equation 2, as it does not influence the gradient computation.

---

### Decision · Action_Editor_zF61 · 2026-02-26

**Recommendation:** Accept as is

**Audience:**

Yes

**Audience Explanation:**

All reviewers acknowledge that the paper will be interesting to TMLR audience.
Reviewer CWyY made the following comment: RAG systems are currently ubiquitous in deployed LLM applications. As the field moves toward Multimodal/Visual RAG to handle complex documents (PDFs with charts, figures), understanding the security surface of these new architectures is critical. This paper is timely as it highlights a specific vulnerability (visual poisoning) that does not exist in text-only pipelines, directly interesting researchers in adversarial ML, multimodal learning, and safety/security.

**Claims And Evidence:**

Yes

**Claims Explanation:**

All the reviewers are satisfied that the claims in the submission are supported by convincing evidence.
Reviewer TsaN stated that the paper shows that VD-RAG systems can be poisoned with a single image, especially under white-box settings, and that the generation component can be very sensitive once a malicious image is in context. The paper also shows that universal poisoning is significantly harder and largely limited to CLIP-like retrieval modules.